# Antibody Response of Combination of BNT162b2 and CoronaVac Platforms of COVID-19 Vaccines against Omicron Variant

**DOI:** 10.3390/vaccines10020160

**Published:** 2022-01-21

**Authors:** Ka-Wa Khong, Danlei Liu, Ka-Yi Leung, Lu Lu, Hoi-Yan Lam, Linlei Chen, Pui-Chun Chan, Ho-Ming Lam, Xiaochun Xie, Ruiqi Zhang, Yujing Fan, Kelvin Kai-Wang To, Honglin Chen, Kwok-Yung Yuen, Kwok-Hung Chan, Ivan Fan-Ngai Hung

**Affiliations:** 1Department of Medicine, Li Ka Shing Faculty of Medicine, University of Hong Kong, Hong Kong, China; kwkhong@connect.hku.hk (K.-W.K.); danlei6@connect.hku.hk (D.L.); mbbs18@hku.hk (H.-M.L.); dachun@connect.hku.hk (X.X.); zhangrq@hku.hk (R.Z.); jyjfan@connect.hku.hk (Y.F.); 2Department of Microbiology, Li Ka Shing Faculty of Medicine, University of Hong Kong, Hong Kong, China; joy2u@connect.hku.hk (K.-Y.L.); u3003963@connect.hku.hk (L.L.); nayiohmal@gmail.com (H.-Y.L.); u3006707@connect.hku.hk (L.C.); bcpc@hku.hk (P.-C.C.); kelvinto@hku.hk (K.K.-W.T.); hlchen@hku.hk (H.C.); kyyuen@hku.hk (K.-Y.Y.); chankh2@hku.hk (K.-H.C.); 3State Key Laboratory for Emerging Infectious Diseases, Li Ka Shing Faculty of Medicine, University of Hong Kong, Hong Kong, China; 4Carol Yu Centre for Infection, Li Ka Shing Faculty of Medicine, University of Hong Kong, Hong Kong, China

**Keywords:** omicron variant, COVID-19, vaccines

## Abstract

By vaccinating SARS-CoV-2 naïve individuals who have already received two doses of COVID-19 vaccines, we aimed to investigate whether a heterologous prime-boost strategy, using vaccines of different platforms as the booster dose, can enhance the immune response against SARS-CoV-2 virus variants. Participants were assigned into four groups, each receiving different combination of vaccinations: two doses of BNT162b2 followed by one dose of BNT162b2 booster (B-B-B); Combination of BNT162b2 (first dose) and CoronaVac (second dose) followed by one dose of BNT162b2 booster (B-C-B); two doses of CoronaVac followed by one dose of CoronaVac booster (C-C-C); two doses of CoronaVac followed by one dose of BNT162b2 booster (C-C-B). The neutralizing antibody in sera against the virus was determined with live virus microneutralization assay (vMN). The B-B-B group and C-C-B group demonstrated significantly higher immunogenicity against SARS-CoV-2 Wild type (WT), Beta variant (BV) and Delta variant (DV). In addition, the B-B-B group and C-C-B group showed reduced but existing protection against Omicron variant (OV). Moreover, A persistent rise in vMN titre against OV was observed 3 days after booster dose. Regarding safety, a heterologous prime-boost vaccine strategy is well tolerated. In this study, it was demonstrated that using vaccines of different platforms as booster dose can enhance protection against SARS-CoV-2 variants, offering potent neutralizing activity against wild-type virus (WT), Beta variant (BV), Delta variant (DV) and some protection against the Omicron variant (OV). In addition, a booster mRNA vaccine results in a more potent immune response than inactivated vaccine regardless of which platform was used for prime doses.

## 1. Introduction

Coronavirus disease 2019 (COVID-19) is an upper respiratory tract infection caused by SARS-CoV-2 and has disrupted our daily lives since the end of 2019, leading to unprecedented global vaccination plans aiming to end the pandemic. Different vaccine platforms were approved for large-scale vaccination, such as inactivated virus vaccines (CoronaVac), viral vector vaccines (ChAdOx1nCoV-19 vaccine), mRNA vaccines (BNT162b2 vaccine), subunit vaccines (S-Trimer vaccine), etc. [1,2,3,4]. With the emergence of the Omicron Variant (OV), vaccine breakthrough is a major concern. It was predicted that Omicron may be twice as likely to escape the current vaccine than the Delta variant (DV) [5], and a previous study conducted by our team demonstrated that the Omicron variant escapes neutralizing antibodies elicited by BNT162b2 or CoronaVac, which is worrying as none of the CoronaVac recipients had a detectable neutralizing antibody against OV, and the seroprotective rate among BNT162b2 recipients was below 25% [6]. The knowledge of whether a third booster dose can rescue the level of neutralizing antibodies against OV is vital in the global endeavor to end the pandemic.

As the current available vaccines were developed based on SARS-CoV-2 wild-type virus (WT), a heterologous prime-boost approach, using different combinations of COVID-19 vaccine candidates, was proposed. Such an approach was demonstrated to be effective in animal models and humans, and different combinations of vaccine platforms and sequences of the combinations demonstrated different effectiveness [7,8,9]. It is therefore hypothesized that using a combination of vaccine platforms, which present the same antigen of SARS-CoV-2 WT with different vector and adjuvants, may enhance protection against virus variants such as OV. The knowledge of safety and immunogenicity against SARS-CoV-2 variants using a heterologous prime-boost approach will aid policy-making such as delivery of vaccines.

## 2. Materials and Methods

### 2.1. Study Design and Participants

This is a prospective cohort study performed in the Hong Kong West Cluster Hospitals under the Hospital Authority in Hong Kong. Twenty-three SARS-CoV-2 naïve healthy individuals who completed two doses of COVID-19 vaccine for at least 6 months were recruited and were given a booster vaccine dose (third dose) according to their preference. Their blood samples were collected before booster vaccination (baseline) and at least 3 days after the booster dose. In addition, 14 participants who had already received the booster dose for more than 3 days were also recruited and had their blood collected (Figure 1). All 37 recruited participants had no known history of COVID-19 infection. The study was approved by the institutional review board of the University of Hong Kong and Hospital Authority (UW 21-214).

### 2.2. Procedure

The nurse administered the vaccines as an intramuscular injection according to the participant’s choice. Recruited participants were then assigned to 4 groups based on the vaccine platforms of their prime dose and booster dose: participants primed with 2 doses of BNT162b2 and received 1 booster dose of IM BNT162b2 (0.3 mL) (B-B-B); participants primed with BNT162b2 (first dose) and CoronaVac (second dose) and received 1 booster dose of IM BNT162b2 (0.3 mL) (B-C-B); participants primed with 2 doses of CoronaVac and received 1 booster dose of IM CoronaVac (0.5 mL) (C-C-C); participants primed with 2 doses of CoronaVac and received 1 booster dose of IM BNT162b2 (0.3 mL) (C-C-B).

CoronaVac (SinoVac Life Sciences, Beijing, China) is a purified inactivated SARS-CoV-2 vaccine candidate developed with CN2 strain of SARS-CoV-2 [10]. BNT162b2 is a mRNA vaccine candidates which encodes SARS-CoV-2 full-length spike protein [1]. Both vaccine candidates used in prime dose and booster dose were developed against the wild-type (WT) strain.

Blood was taken from the participants before the booster dose (baseline) and at least 3 days post-vaccination for the antibody assay. As described by our previous study, live virus microneutralization assay (vMN) was performed in the Biosafety level 3 facility of HKU to determine the level of neutralizing antibody in sera [11]. Serial 2-fold dilutions of serum starting from 1:10 were incubated with 100 median tissue culture infectious doses (TCID50) or SARS-CoV-2 HKU-001a (wild type, GenBank accession number MT230904) strain (WT) [12], Beta variant (BV) (GISAID accession number: EPI_ISL_2423556), Delta variant (DV) (GISAID accession number: EPI_ISL_3221329) and Omicron variant (OV) (hCoV-19/Hong Kong/HKU-691/2021) [6] for 1.5 h at 37 °C. Then, a serum–virus mixture was added to VeroE6/TMPRSS2 cells (JCRB Cell Bank Catalogue no. JCRB1819) on 96-well plates [13]. After 72 h of incubation at 37 °C and 5%CO_2_, the cytopathic effect (CPE) was examined and the antibody titre was determined by the highest dilution with 50% inhibition of CPE (Figure 1).

To assess the safety and adverse events of the booster dose, participants were asked to record any adverse events for 7 days after vaccination.

### 2.3. Outcome

The primary endpoint of this study was the vMN geometric mean titre (GMT) against WT, BV, DV and OV. The secondary endpoints were GMT fold increase and safety. For safety, severe adverse events (SAEs) were defined as death, disabling or life-threatening conditions related to vaccine; adverse events include fever (>38 °C), chills, headache, tiredness, nausea, vomit, diarrhea, muscle pain, joint pain, facial dropping, skin rash or injection site reactions (pain, redness, swelling, ecchymoses, itching).

### 2.4. Statistical Analysis

A statistical inference of normally distributed continuous variables was performed using t-tests and one way ANOVA, including demographic parameters (age), GMT and GMT fold increase. Categorical variables were analyzed using Pearson’s chi-squared test and Fisher’s exact test. When *p* < 0.05, the result was statistically significant. SPSS statistics (IBM Corp. Released 2020. IBM SPSS Statistics for Macintosh, Version 27.0. Armonk, NY: IBM Corp) and GraphPad PRISM 9(Version 9.3.1, for macOS, GraphPad Software, San Diego, CA, USA, www.graphpad.com) were used for statistical computation.

## 3. Results

### 3.1. Subjects

Between November 2021 and December 2021, 37 SARS-CoV-2 naïve individuals who already received two doses of COVID-19 vaccine for at least 6 months or have recently received the third booster dose were recruited to the study. Recruited participants were assigned into four groups depending on the combination of vaccines they received: participants primed with two doses of BNT162b2 and one booster dose of BNT162b2 (B-B-B, *n* = 15, median age = 53 years); participants primed with BNT162b2 (first dose) and CoronaVac (second dose) who received one booster dose of BNT162b2 (B-C-B, *n* = 5, median age = 47 years); participants primed with two doses of CoronaVac who received one booster dose of CoronaVac (C-C-C, *n* = 9, median age = 58 years); participants primed with two doses of CoronaVac and who received one booster dose of BNT162b2 (C-C-B, *n* = 8, median age = 58.5 years). There was no statistically significant difference in age (*p* = 0.54), sex ratio (*p* = 0.888), comorbidities (*p* = 0.395) and number of days after the booster dose when the blood was sampled (*p* = 0.244) between all the groups (Table 1)

### 3.2. Immunogenicity of Different Vaccine Combinations

The level of neutralizing antibody (nAb) in sera was determined by vMN. For SARS-CoV-2 wild type (WT) virus, sera from participants in the B-B-B group had a significantly higher level of antibody than the other groups (*p* = 0.046), after administration of the booster dose, B-B-B group (306, 95% CI, 154–608) and C-C-B group (207, 95% CI, 22.7–1893) had a significant higher Geometric mean titre (GMT) than C-C-C group (34.3, 95% CI, 16.3–72.1) (Figure 2a). The GMT fold increase was significantly higher in the B-B-B group (15.3, 95% CI, 7.14–32.7) and C-C-B group (36.1, 95% CI, 4.21–310). GMT level is also boosted in B-C-B group but is not significantly higher than C-C-C group (Table 2).

For vMN titre against Beta variant (BV), there was no significant difference in the baseline GMT level across the groups. After administration of booster dose, B-B-B group (175, 95% CI, 95–324) had a significant higher GMT level than C-C-C group (18.5, 95% CI, 11.3–30.4). The GMT level of B-C-B group (106, 95% CI, 25–446), C-C-B group (87.2, 95% CI, 14.5–523) was also elevated but was not significantly higher than the GMT level of the C-C-C group (Figure 2b). The GMT fold increase was also higher in the B-B-B group (9.4, 95% CI, 5.77–15.3), B-C-B group (21.1, 95% CI, 5–89.1) and C-C-B group (17.4, 95% CI, 2.91–105) (Table 2).

Regarding immunity against the Delta variant (DV), there was some pre-existing nAb against the virus in sera of the B-B-B and C-C-C groups. After booster dose, the B-B-B group (184, 95% CI, 81.7–413) and C-C-B group (160, 95% CI, 17.5–1461) showed a significantly higher level of GMT than the C-C-C group (20, 95% CI, 11.7–34.1). There was also an elevated GMT level in the B-C-B group (139, 95% CI, 21.6–900) but it was not significantly higher than that of the C-C-C group (Figure 2c). The GMT fold increase was also higher for B-B-B group (13.9, 95% CI, 5.75–33.7), B-C-B group (27.9, 95% CI, 4.31–180) and the C-C-B group (32, 95% CI, 3.5–292) (Table 2).

The immunogenicity of the booster dose against the Omicron variant (OV) is markedly reduced. For immunity against Omicron variant (OV), there was no pre-existing immunity in any of the group. After the booster dose, the GMT level was relatively higher in the B-B-B group (27.6, 95% CI, 15–51) and the C-C-B group (23.8, 95% CI, 6.45–87.7) than the C-C-C group (5.83, 95% CI, 4.61–7.38) and the B-C-B group (10, 95% CI, 2.25–44.4) (Figure 2d), but the GMT levels were non-comparable to the GMT levels against other variants. The 95% confidence interval of GMT levels from B-C-B group and C-C-C group overlaps with the baseline and is therefore considered non-significant. For GMT fold increase, B-B-B group (5.53, 95% CI, 2.99–10.2) and C-C-B group (4.76, 95% CI, 1.29–17.5) was shown to have some increase (*p* = 0.077) but not the B-C-B group and C-C-C group (Table 2).

### 3.3. Changes in GMT against OV Level after Booster Dose

In terms of changes in the vMN titre against the Omicron variant after booster dose, the GMT titre is 5 (95% CI, 5–5) at baseline, which raised to 9.52 (95% CI 4.76–19) at day 3–7, 26.4 (95% CI, 10.3–67.6) at day 8–14 and 23.5 (95% CI, 11.6–47.7) after day 14. Surprisingly, the GMT titre after day 14 was similar to that at day 8–14 (Figure 3). This may be due to small sample size and difference in participants characteristics, as participants included in the >D14 group is significantly older (Appendix A).

### 3.4. Safety

Pain at the site of injection was the most common adverse events and there was no statistically significant difference between all the groups (*p* = 0.733). Other injection site reactions such as redness (*p* = 0.523), swelling (*p* = 0.560) were also reported. Interestingly, injection site itchiness was reported and was significantly higher in the B-C-B group (*p* = 0.037). Other common adverse events reported include headache (*p* = 0.063), tiredness (*p* = 0.763), muscle pain (*p* = 0.899) and joint pain (*p* = 0.483). Fever (*p* = 0.174), chills (*p* = 0.196) and diarrhea (*p* = 0.103) were reported in a few cases. There was no report of severe adverse events (Table 3).

## 4. Discussion

Recent emergence of the Omicron variant of SARS-CoV-2 has raised great concern about vaccine efficacy, as evidenced by reduced neutralization titers from sera collected from COVID-19 vaccine recipients [6], leading to a call for updated vaccines and booster doses [14,15]. However, it takes time for new vaccines to be developed. Therefore, enhancing immunity against the Omicron variant using currently approved vaccines is critical to controlling the casualty caused by this new variant of concern (VOC).

A heterologous prime-boost vaccine strategy was adopted in vaccines against different pathogens, such as HIV. It was suggested that a heterologous prime-boost strategy is more immunogenic than homologous prime boost [16]. For COVID-19 vaccines, a recently published review paper concludes that such strategy using vaccines of different platforms shows improved immunogenicity and flexibility profiles for future vaccination in time of global shortage of vaccines [17]. The immunogenicity and safety of booster dose was studied in individuals who received two doses of ChAdOx1 nCov-19 or BNT162b2 vaccines and was shown to be effective in boosting antibody and neutralizing responses with no safety concern [9]. Regarding vaccine-induced immunity against the Omicron variant, it was demonstrated that individuals who received two doses of inactivated vaccine (CoronaVac) can benefit both from a booster dose of heterologous protein subunit vaccine and a booster dose of homologous inactivated vaccine [18].

Our team have previously demonstrated that for SARS-CoV-2 naïve individuals primed with two doses of inactivated vaccines (CoronaVac), a booster dose of mRNA vaccine (BNT162b2) offers more potent neutralizing activity against the Delta variant compared to using inactivated vaccine as a booster dose [19]. Similarly, our current study shows that a heterologous prime-boost strategy with mRNA vaccine as the booster dose in individuals previously primed with inactivated vaccines can induce a more potent immune response against SARS-CoV-2 variants, including OV. A mRNA booster dose can also offer protection against SARS-CoV-2 variants in individuals primed with mRNA vaccines. One possible explanation could be that mRNA vaccines lead to more focused CD4 and CD8 T-cells stimulation against the spike protein, while inactivated vaccine induces a response that is more diffuse, targeting multiple different proteins, as BNT162b2 vaccine presented the spike proteins as the only antigen while CoronaVac presented the whole virus [1,10,20]. A local study from Hong Kong demonstrated that mRNA vaccine induced a higher level of neutralizing antibody against SARS-CoV-2 compared to inactivated vaccine [21]. In addition, there were also studies suggesting that BNT162b2 mRNA vaccine can induce some broad cross-reactivity antibodies [22]. Therefore, it is proposed that mRNA vaccines can be considered for booster dose to better enhance the neutralizing activity against Omicron variant.

As shown in this study, individuals who received two doses of inactivated vaccine and an inactivated booster dose had a less potent immune response against virus variants; therefore, these individuals may consider receiving a mRNA booster dose for better protection against virus variants. In terms of safety, our study shows that there were no statistically significant differences in terms of side effects between different combinations of vaccines except for itchiness, suggesting that a heterologous prime-boost vaccines strategy has a similar safety profile compared to a homologous prime-boost vaccine strategy, and is well tolerated.

The limitation of this study is the small sample size, which lead to wide 95% confidence intervals of GMT. In addition, some of the patients were recruited after they received the booster dose and their baseline neutralizing antibody level before vaccination was lacking. For future work, cellular immunity induced by a heterologous prime-boost strategy should be evaluated, and the recruited patients should also be followed up to monitor the changes in neutralizing antibody level in long term.

## 5. Conclusions

Our study found that a heterologous prime-boost approach using one booster dose of mRNA vaccine (booster dose) can enhance protection against SARS-CoV-2 variants, including wild type, Beta variant, Delta variant and Omicron variant. Although the response against Omicron variant (OV) is less potent compared to other variants, recipients of the third booster dose vaccine can still benefit from it. In conclusion, our study demonstrated that combination of vaccine platforms can be a potential vaccine strategy against emergence of virus variants.

## Figures and Tables

**Figure 1 vaccines-10-00160-f001:**
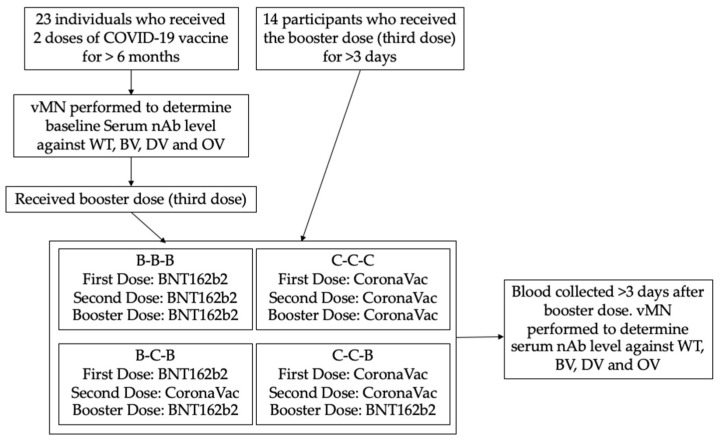
Procedure of the study. vMN: virus microneutralization assay; nAb: neutralizing antibody; WT: SARS-CoV-2 wild type; BV: SARS-CoV-2 Beta variant; DV: SARS-CoV-2 Delta variant; OV: SARS-CoV-2 Omicron variant.

**Figure 2 vaccines-10-00160-f002:**
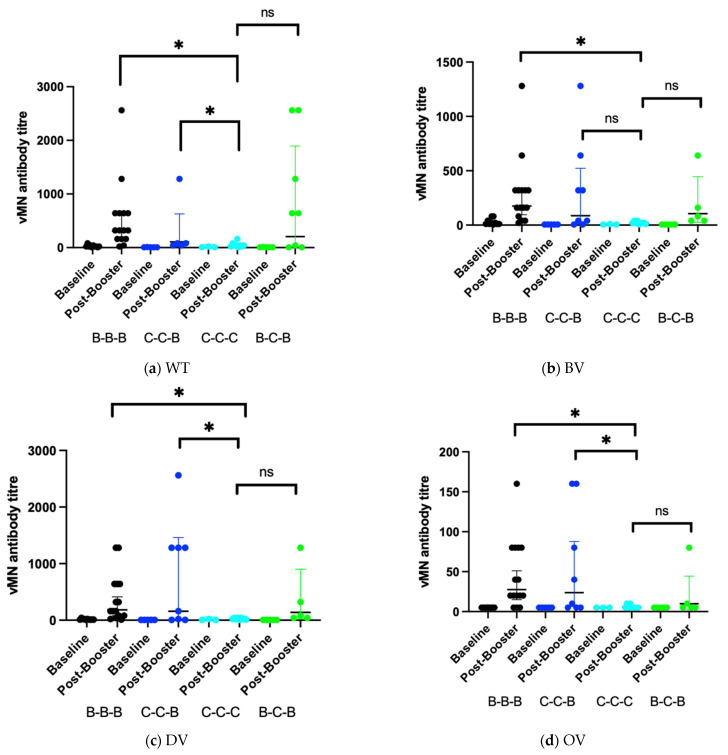
GMT titre of different vaccine combinations against SARS-CoV-2 variants at baseline and post booster vaccination. (**a**) Wild type; (**b**) Beta variant; (**c**) Delta variant; (**d**) Omicron variant; * = *p* < 0.05; ns = non-significant.

**Figure 3 vaccines-10-00160-f003:**
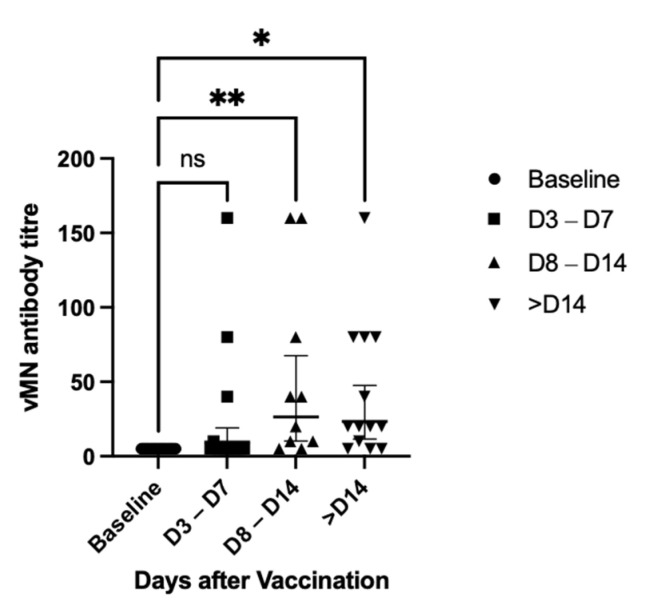
vMN titre against Omicron variant after booster dose, all groups included; * = *p* < 0.05; ** = *p* < 0.005; ns = non-significant.

**Table 1 vaccines-10-00160-t001:** Baseline characteristics of subjects.

	B-B-B (*n* = 15)	B-C-B (*n* = 5)	C-C-C (*n* = 9)	C-C-B (*n* = 8)	*p*-Value
Age (Years)	53 (26–76)	47 (22–58)	58 (31–64)	58.5 (27–70)	0.54
Sex					0.888
Male	7 (46.7%)	2 (40%)	5 (55.6%)	3 (37.5%)	
Female	8 (53.3%)	3 (60%)	4 (44.4%)	5 (62.5%)	
Comorbidities	4 (26.7%)	1 (20%)	5 (55.6%)	2 (25%)	0.395
Date Post-3rd Dose (Days)	14 (3–39)	5 (5–7)	14 (5–32)	8 (4–29)	0.2443

Data are median age (range) or *n* (%); B-B-B: participants primed with 2 doses of BNT162b2 and 1 booster dose of BNT162b2; B-C-B: participants primed with BNT162b2 (first dose) and CoronaVac (second dose), and received 1 booster dose of BNT162b2; C-C-C: participants primed with 2 doses of CoronaVac and received 1 booster dose of CoronaVac; C-C-B: participants primed with 2 doses of CoronaVac and received 1 booster dose of BNT162b2; Comorbidities: hypertension (HT), ischemic heart disease (IHD), diabetes mellitus (DM), stroke, chronic heart failure (CHF), malignancy, asthma, chronic obstructive pulmonary disease (COPD) and thyroid diseases. Median number of days post-third-dose vaccination (range).

**Table 2 vaccines-10-00160-t002:** Immunogenicity of different vaccine platform.

	B-B-B (*n* = 15)	B-C-B (*n* = 5)	C-C-C (*n* = 9)	C-C-B (*n* = 8)	*p*-Value
WT					
Baseline GMT ^1^	20 (10.8–37.1) ^2^	5.74 (3.91–8.44)	7.94 (1.09–58) ^3^	5.74 (3.91–8.44) ^4^	0.046
Post-Booster GMT	306 (154–608)	106 (17.7–629)	34.3 (16.3–72.1)	207 (22.7–1893)	0.058
GMT fold increase	15.3 (7.14–32.7) ^5^	18.4 (2.84–119)	4.32 (1.99–9.37) ^5^	36.1 (4.21–310) ^5^	0.012
BV					
Baseline GMT	18.7 (9.46–36.8) ^2^	5 (5–5)	6.3 (2.33–17) ^3^	5 (5–5) ^4^	0.086
Post-Booster GMT	175 (95–324)	106 (25–446)	18.5 (11.3–30.4)	87.2 (14.5–523)	0.148
GMT fold increase	9.4 (5.77–15.3) ^5^	21.1 (5–89.1)	2.94 (1.84–4.7) ^5^	17.4 (2.91–105) ^5^	0.026
DV					
Baseline GMT	13.2 (8.69–20) ^2^	5 (5–5)	7.94 (1.09–58) ^3^	5 (5–5) ^4^	0.044
Post-Booster GMT	184 (81.7–413)	139 (21.9–900)	20 (11.7–34.1)	160 (17.5–1461)	0.041
GMT fold increase	13.9 (5.75–33.7) ^5^	27.9 (4.31–180)	2.52 (1.36–4.66) ^5^	32 (3.5–292) ^5^	0.011
OV					
Baseline GMT	5 (5–5) ^2^	5 (5–5)	5 (5–5) ^3^	5 (5–5) ^4^	-
Post-Booster GMT	27.6 (15–51)	10 (2.25–44.4)	5.83 (4.61–7.38)	23.8 (6.45–87.7)	0.077
GMT fold increase	5.53 (2.99–10.2) ^5^	2 (0.45–8.88)	1.17 (0.992–1.42) ^5^	4.76 (1.29–17.5) ^5^	0.077

^1^: data are mean value (95% CI); ^2^: *n* = 10, 10 participants recruited before booster dose; ^3^: *n* = 3, 3 participants recruited before booster dose; ^4^: *n* = 5, 5 participants recruited before booster dose; ^5^: baseline of participants without baseline is assumed to be the same as mean of those in the same group against that particular virus variant.

**Table 3 vaccines-10-00160-t003:** Adverse events.

	B-B-B (*n* = 15)	C-C-B (*n* = 6)	C-C-C (*n* = 9)	B-C-B (*n* = 5)	*p*-Value
Fever	0 (0%)	1 (16.7%)	0 (0%)	0 (0%)	0.174
Chills	0 (0%)	1 (16.7%)	0 (0%)	1 (20%)	0.196
Headache	2 (13.3%)	3 (50%)	0 (0%)	2 (40%)	0.063
Tiredness	6 (40%)	3 (50%)	4 (44.4%)	1 (20%)	0.763
Nausea	0 (0%)	0 (0%)	0 (0%)	0 (0%)	-
Vomiting	0 (0%)	0 (0%)	0 (0%)	0 (0%)	-
Diarrhea	0 (0%)	0 (0%)	0 (0%)	1 (20%)	0.103
Muscle Pain	4 (26.7%)	2 (33.3%)	2 (22.2%)	2 (40%)	0.899
Joint Pain	2 (13.3%)	2 (33.3%)	2 (22.2%)	0 (0%)	0.483
Facial Dropping	0 (0%)	0 (0%)	0 (0%)	0 (0%)	-
Skin Rash	0 (0%)	0 (0%)	0 (0%)	0 (0%)	-
SAE ^1^	0 (0%)	0 (0%)	0 (0%)	0 (0%)	-
Injection Site Reaction					
Pain	12 (80%)	5 (83.3%)	7 (77.8%)	5 (100%)	0.733
Redness	1 (6.67%)	1 (16.7%)	0 (0%)	1 (20%)	0.523
Swelling	3 (20%)	1 (16.7%)	0 (0%)	1 (20%)	0.560
Ecchymoses	0 (0%)	0 (0%)	0 (0%)	0 (0%)	-
Itching	2 (13.3%)	1 (16.7%)	0 (0%)	3 (60%)	0.037

^1^: SAE: Severe adverse event, vaccine-related undesired events including death, disability or life-threatening conditions.

## Data Availability

The data used to support the findings of this study are included within the article.

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
