# Peer review of "Antibody Response of Combination of BNT162b2 and CoronaVac Platforms of COVID-19 Vaccines against Omicron Variant"

_vaccines, 2022, doi:10.3390/vaccines10020160_

Round 1
Reviewer 1 Report
Today, when a new Omicron variant of SARS-CoV-2 has appeared in circulation, and vaccines against the gene are not yet ready, such a study seems to be very important and inspiring. This is very useful and timely research, which was carried out with high quality. However, a revision of the manuscript is needed and a number of modifications are required.
Point 1: The authors constantly talk about the heterologous prime-boost approach and heterologous prime-boost strategy which they used but they do not indicate what this heterology is - in different platforms, different strains, both? Please indicate against which strains the tested vaccines were designed and with which antigenic variants the neutralization test was performed. For clarity, it would be great to summarize this as a table in the Materials and Methods section.
Point 2: Line 44-46. Please, briefly describe not only the main known platforms which are currently in use in the development of COVID-19 vaccines but provide appropriate examples.
Point 3: Line 221-225. For the first time, the authors explained that the BNT162b2 vaccine they use is based on mRNA technology, while CoronaVac is a conventional inactivated vaccine almost only at the very end of the article. Please add this explanation to the Materials and Methods section, preferably in the proposed table (see point 1). Please specify there that the first vaccine is based on mRNA technology, while the second is a conventional inactivated vaccine and against what particular variants of SARS-CoV-2 there were designed.
Point 4: Figure 1 and 2. Please, decipher the abbreviation “ns”
Point 5: Please discuss in the Discussion section the possible mechanism of the boosting effect of the mRNA vaccine in more detail. Why does a vaccine that is considered by many researchers not the most immunogenic provide the best boosting effect? The authors suggested, “One possible explanation could be that mRNA vaccines lead to a more focus CD4 and CD8 T-cells stimulation against the spike protein.” Why do the BNT162b2-elicited sera efficiently neutralize diverse SARS-CoV-2 variants?
Point 6: Line 35, 43, 206, 226, 251. Please, delete the word “virus” from combinations of words “SARS-CoV-2 virus” because the letter “V” already means “virus.” This is a tautology.
Point 7: Line 223. Please, add a dot after ref. (14).
Reviewer 2 Report
The paper describes a study done using 4 different combinations of vaccine - 3 doses ( 2 doses plus booster) to evaluate the antibody response in the light of Omicron variant.
A very useful study and well explained in the paper. I dont consider it a weakness but the only point I commented on is that ( not usre whether I missed it) is that the inclusion criteria should also state that the persons included in the study either did not have COVID recently and were currently PCR negative when the booster dose was being given. This will ensure that there is no confusion whether the antibody response was due to COVID infection or the booster dose.
I would only reiterate somewhere in the method section that all the sample cases had not only received 2 doses of vaccine earlier but also did not have COVID and were COVID negative as part of inclusion criteria
Author Response
RE: A point-by-point response to reviewer’s comment
“Antibody Response of combination of BNT162b2 and CoronaVac Platforms of COVID-19 vaccines against Omicron Variant”
By Ka-Wa Khong, Danlei Liu, Ka-Yi Leung, Lu Lu, Hoi-Yan Lam, Linlei Chen, Pui-Chun Chan, Ho-Ming Lam, Xiaochun Xie, Ruiqi Zhang, Yujing Fan, Kelvin Kai-Wang To, Honglin Chen, Kwok-Yung Yuen, Kwok-Hung Chan and Ivan Fan-Ngai Hung*
We are grateful to the comments and suggestions given by editors and referees and have revised the manuscript accordingly. A point-to point response to reviewer’s comment is given as follow:
Reviewer 2
The paper describes a study done using 4 different combinations of vaccine - 3 doses ( 2 doses plus booster) to evaluate the antibody response in the light of Omicron variant.
A very useful study and well explained in the paper. I dont consider it a weakness but the only point I commented on is that ( not usre whether I missed it) is that the inclusion criteria should also state that the persons included in the study either did not have COVID recently and were currently PCR negative when the booster dose was being given. This will ensure that there is no confusion whether the antibody response was due to COVID infection or the booster dose.
I would only reiterate somewhere in the method section that all the sample cases had not only received 2 doses of vaccine earlier but also did not have COVID and were COVID negative as part of inclusion criteria
Response: We thank Reviewer 2 for the comment. We have clarify the inclusion criteria accordingly.
Line 76-77: All 37 recruited participants had no known history of COVID-19 infection.
Because of low prevalence of COVID-19 infection in Hong Kong, PCR for SARS-CoV-2 was not performed during recruitment. Participants who had no record of COVID-19 infection were assumed to be SARS-CoV-2 naïve.
